# Amino Acid-Based Natural Deep Eutectic Solvents for Extraction of Phenolic Compounds from Aqueous Environments

**Meiyu Li** [1], **Yize Liu** [1], **Fanjie Hu** [1], **Hongwei Ren** [1,2,3,*] and **Erhong Duan** [1,2,3,*]

[1] School of Environmental Science and Engineering, Hebei University of Science and Technology, Shijiazhuang 050018, China; lmy15103273811@hotmail.com (M.L.); LiuYizee@hotmail.com (Y.L.); hufanjie964831547@hotmail.com (F.H.)

[2] Pollution Prevention Biotechnology Laboratory of Hebei Province, Shijiazhuang 050018, China

[3] National-Local Joint Engineering, Center of Volatile Organic Compounds & Odorous Pollution Control Technology, Shijiazhuang 050018, China

\* Correspondence: renhongweirhw@hebust.edu.cn (H.R.); deh@hebust.edu.cn (E.D.); Tel./Fax: +86-10-8973-3335 (E.D.)

**Abstract:** The environmental pollution of phenol-containing wastewater is an urgent problem with industrial development. Natural deep eutectic solvents provide an environmentally friendly alternation for the solvent extraction of phenol. This study synthesized a series of natural deep eutectic solvents with L-proline and decanoic acid as precursors, characterized by in situ infrared spectrometry, Fourier transform infrared spectrometry, hydrogen nuclear magnetic resonance spectrometry, and differential thermogravimetric analysis. Natural deep eutectic solvents have good thermal stability. The high-efficiency extraction of phenol from wastewater by natural deep eutectic solvents was investigated under mild conditions. The effects of natural deep eutectic solvents, phenol concentration, reaction temperature, and reaction time on phenol extraction were studied. The optimized extraction conditions of phenol with L-prolin/decanoic acid were as follows: molar ratio, 4.2:1; reaction time, 60 min; and temperature, 50 °C. Extraction efficiency was up to 62%. The number of extraction cycles can be up to 6, and extraction rate not less than 57%. The promising results demonstrate that natural deep eutectic solvents are efficient in the field of phenolic compound extraction in wastewater.

**Keywords:** amino acid; natural deep eutectic solvents; phenolic wastewater





## 1. Introduction

As a class of protoplasmic toxicants, phenolic compounds have the characteristics of numerous sources, high toxicity, and difficult degradation [1–3]. Vast amounts of phenol and phenolic compounds are produced by industry, including textiles, ceramics, medicine, coking, and leather processing, and then discharged into the environment, producing lasting adverse effects. Phenolic compounds are a kind of poison, such as phenol; they can poison almost all living things, and are particularly harmful to the human nervous system [4–6]. Therefore, many countries and regions strictly restrict the discharge of phenolic wastewater [7]. For instance, according to standards for drinking water quality, the maximal permissible concentration of volatile phenols in aqueous solution is 0.002 mg L$^{-1}$ in China, and 0.005 mg L$^{-1}$ in Japan. In addition, the maximal accepted concentration of phenol in wastewater is 1 mg L$^{-1}$ under European Union rules [8].

The main methods for the disposal of phenolic wastewater can be divided into two types: destructive and recyclable methods [9–11]. Destructive methods may be thermal decomposition, catalytic degradation, and chemical oxidation, whereas recyclable methods include solvent extraction, adsorption, and membrane separation. Recyclable methods are usually adopted when phenol content is at or over 1000 mg L$^{-1}$. Among them, solvent extraction is a preferred recyclable treatment method, especially for high concentrations of phenolic wastewater (e.g., sodium sulfate wastewater), due to it having excellent properties such as high selectivity, easy availability, and large extraction capacity. The type of

extractant is a major factor during extraction. In the last few decades, common traditional extractants have many problems, such as toxicity, expensiveness, and low extraction yield. Aiming at the above problems, researchers are looking for a novel type of extractant that is cheap, accessible, environmentally friendly, and highly extractive to replace traditional extractants. In recent years, ionic liquids (ILs), natural deep eutectic solvents (NADESs), and biosolvents (Bio-Ss), as novel alternatives, have emerged as promising candidates in this field of research [12–14].

Abbott's group conducted some interesting experiments at the beginning of our century [15,16]. A few published papers of this group triggered research in the area of deep eutectic solvents (DESs). They are generally synthesized by the simple eutectics of hydrogen-bond donors (HBDs) and hydrogen-bond acceptors (HBAs) from components of natural origin such as alcohol, organic acids, and amino acids; thus, they are also defined as natural deep eutectic solvents (NADESs) [17,18].

Compared to ILs, DES have superior biocompatibility and high reactivity [19–21], and are called novel ionic liquids. In the past few years, NADESs have been widely applied as extractants, catalysts, electrolytes, and solvents. Li et al. [22] expounded the synthesis, application, and development of deep eutectic solvents, indicating that it is expected to diversify into extraction and separation. Afterwards, Zhao et al. [23] focused on aqueous two-phase systems based on deep eutectic solvents and their application in green separation processes. Lalit Khare et al. [24] synthesized hydrophobic DES using menthol, thymol, or tetra butyl ammonium hydrogen bromide as HBA, and various long-chain acids as HBD, for extracting ergosterol from mushrooms. The experiment used response surface methodology (RSM) and obtained a maximal extraction yield of 6995.00 μg ergosterol/g dry weight. In 2018, Zhang et al. [25] prepared amino acid ionic liquid-based deep eutectic solvents (tryptophan fluoborate/urea, TrpBF4/U) from L-tryptophan, fluoroboric acid, and urea for oil–carbonate mineral separation. Results showed that oil recovery was enhanced by at least 11% compared with the initial status. In 2019, A Shishov et al. [26] developed in situ deep eutectic mixtures supported in a hydrophilic porous membrane. By hydrogen–bond interaction, the analytes were extracted and retained into the hydrophilic porous membrane. Lastly, deep eutectic mixtures were applied to the HPLC-FLD determination of phenols in smoked food samples. In addition to the extraction of phenolic compounds from food, there is also the extraction of phenolic compounds from wastewater. According to the investigation of Yang [27], a ternary extractant consisting of 20% tributylphosphane (TBP)/20% diethyl carbonate (DEC)/60% cyclohexane was utilized to extract phenol from wastewater, of which the maximal extraction rate could reach 99.79%. Mechanism analysis illustrated that the extraction of phenol was enabled by intermolecular hydrogen bonding with both TBP and DEC. A NaOH solution was utilized in the experimental process, which may have caused secondary pollution. In addition, many researchers utilized the synthesis of different kinds of DES for absorbing and extracting contaminants containing pesticides, radioactive iodine polluting gases, and the like [28]. For their latest study, Tang [29] used green renewable choline chloride as new adsorbents for the removal of aniline by forming choline-based deep eutectic solvents, and this displayed excellent aniline removal efficiency. In subsequent research by Tang [30], a polarity-controlled hexafluoroisopropanol-choline chloride (DES)-based biphasic system was successful developed. Polarity-driven recognition enables it to be an efficient absorbent in the simultaneous extraction and separation of high-polarity compounds.

To sum up, a variety of DESs have been applied in the separation field, and promising results demonstrated DES to be an attractive option for the treatment of hazardous substances from the environment. However, there is still difficulty in the large application of DES for secondary NaOH solution pollution, known high viscosity, and relatively low reactivity [31,32].

This research develops a hydrophobic, low-viscosity, and high-reactivity NADES. The biodegradable amino acid L-proline was determined as HBD. The heterocyclic structure, sufficient carboxyl, and amino groups on L-proline could enable multiple potential reac-

tion sites. Decanoic acid was preferred as the feedstock of HBA. The long hydrophobic chains on decanoic acid ensured the hydrophobicity of NADES, which is beneficial for the extraction of decanoic phenol form wastewater. The decanoic acid/L-proline NADES ensures extraction efficiency, reduces the risk of reaction to a certain extent, and avoids the secondary pollution of related hazardous substances. Amino acid-based NADESs formed with decanoic acid and L-proline in different proportions were developed to realize the efficient extraction of phenolic compounds from aqueous environments. Thermogravimetric analysis (TG) with UV–vis absorption spectra manifested its physical and optical properties. Fourier transform infrared spectroscopy (FTIR) and an in situ infrared spectrometer system were utilized to study the extraction mechanism. In addition, the effects of precursor ratio, phenol concentration in simulated wastewater, reaction time, and temperature on the removal rate of phenol were investigated. The effect of regeneration time on extraction efficiency was also assessed.

## 2. Materials and Methods

### 2.1. Chemicals and Materials

L-proline (purity $\geq$ 98%) was obtained from Sichuan Huatang Jurui Biotechnology Company Limited (Co. Ltd). Decanoic acid (purity $\geq$ 98%) and phenol (purity $\geq$ 98%) were purchased from Shanghai McLean Biochemical Technology Co. Ltd. All agents were used as received and with no further purification. In the experimental process, all aqueous solutions were prepared using high-purity (Milli-Q) water with specific conductance < 0.1 mS cm$^{-1}$.

### 2.2. Preparation of Amino Acid-Based Natural Deep Eutectic Solvents

In the current work, a series of amino acid-based NADESs were prepared from L-proline (L-Pro) and decanoic acid according to previous methods. L-Pro was used as hydrogen bond acceptor, and decanoic acid was used as hydrogen bond donor [33–35]. L-Pro and decanoic acid were added into a 250 mL flask at different molar ratios (decanoic acid:L-Pro, 2.88:1, 3.5:1, 4.2:1, 4.55:1, 4.84:1). Taking 5 °C min$^{-1}$ as the heating rate and starting from 20 °C to increase the temperature, the preparation of amino acid-based NADESs was observed with different proportions. The whole process was carried out in a magnetic stirrer heated by a constant temperature collector. The container was also protected with nitrogen until a uniform, stable liquid. The products were dried in a forced air oven at 60 °C for 24 h.

### 2.3. Extraction-Cycle Procedure

#### 2.3.1. Preparation of Simulated Wastewater

First, 1 g solid pure phenol was weighed and placed in a clean beaker. After adding an appropriate amount of high-purity water until complete dissolution, it was moved to a 1 L volumetric flask with constant volume and shaken well to obtain a phenol-containing wastewater model with a concentration of 1000 mg L$^{-1}$ [36]. The phenol wastewater model with concentrations of 2000, 3000, 4000, and 20,000 mg L$^{-1}$ was obtained by repeating the above operation steps [37]. The above solutions were proportionally diluted to obtain a series of wastewater models with concentration gradients of 0.5, 1, 2, 5, 10, 20, 40, 60, 80, 10,000, 15,000, and 5000 mg L$^{-1}$, respectively.

#### 2.3.2. Extraction Process

The NADES with a ratio of decanoic acid to L-Pro of 4.2:1 was heated to a homogeneous transparent liquid. Then, 5 mL of the NADES and 10 mL of 100 mg L$^{-1}$ phenol simulated wastewater were injected into a three-mouth flask [38]. The mixture was then heated at 50 °C under magnetic stirring with 10 rotations per second continuing for 60 min, and poured into the test tube to make it stand. Taking the subnatant after complete delamination, its absorbance was measured by UV–vis absorption spectra. The corresponding residual phenol concentration was calculated, and the phenol removal rate at that temper-

ature was then obtained. The above operations were then repeated at ratios of decanoic acid to L-Pro of 2.88:1, 3.5:1, 4.55:1, and 4.84:1. In addition, the control variable method was used to explore the influence of phenol concentration, reaction time, and reaction temperature on the removal rate of phenol [39]. The experiment was measured 3 times in parallel.

### 2.4. Analytical Methods

Hydrogen nuclear magnetic spectra ($^1$H NMR) were recorded using a nuclear magnetic resonance spectrometer (Bruker, Avance 500 MHz, Karlsruhe, Germany) at room temperature. The solvent was $D_2O$. Thermogravimetric (TG) (Netzsch, STA449 F5, Nuremberg, Bavaria, Germany) analysis was performed under helium. Samples for TG analysis were placed in a small crucible with a cover under the helium atmosphere with a flow rate of 20 mL min$^{-1}$, and heated at a temperature range of 50–400 °C with a heating rate of 30 K min$^{-1}$. Fourier transform infrared spectra were recorded on a Fourier transform infrared spectrometer (FTIR) (Bruker Tensor 27, Karlsruhe, Germany) with wavenumbers from 4000 to 400 cm$^{-1}$. An in situ infrared spectrometer system (Mettler Toledo, React IR 15, Zurich, Switzerland) can be used for tracing and detecting chemical reactions over the entire course of an experiment in order to analyze the mechanism and process of a reaction. The pH of the sample was determined by a Seven Excellence multiparameter tester from Mettler Toledo. The absorbance of residual phenol after NADES extraction was collected with a UV-6100S UV–vis spectrophotometer (Metash instrument Co., Ltd., Shanghai, China). The scan range was 190 to 1100 nm with an interval of 0.5 nm.

### 2.5. Calculation of Phenol Extraction Rate

The completely reacted mixture was placed in a 10 mm quartz colorimeter to measure its absorbance under UV–visible light. The concentration of phenol in the mixture was calculated by Lambert–Beer's law. The extraction rate of phenol was calculated according to the following formula:

$$R = \frac{C_P - C_{P\text{-NADES}}}{C_P} \times 100\% \tag{1}$$

where $R$, extraction rate of phenol, %; $C_P$, phenol content of model wastewater, mol L$^{-1}$; $C_{P\text{-NADESs}}$, residual phenol content after extraction, mol L$^{-1}$.

## 3. Results and Discussion

### 3.1. Preparation of Natural Deep Eutectic Solvents

The required temperature for the formation of acid-based NADESs in different proportions is different. The required temperature for synthesizing each amino acid-based NADES is shown in Figure 1. The synthesis temperature of amino acid-based NADESs under different ratios (decanoic acid:L-Pro, 2.88:1, 3.5:1, 4.2:1, 4.55:1, 4.84:1) is 60, 50, 40, 45, and 55 °C, respectively. Briefly, the lowest temperature for the synthesis of NADESs by decanoic acid and L-pro is 40 °C, at which point the molar ratio of is 4.2:1.

The heating and stirring process of decanoic acid and L-pro was monitored with online in situ infrared [40]. The peak value at 1700 cm$^{-1}$ significantly and stably decreased with the increase in reaction time from 0 to 60 min (Figure 2). A hydrogen bond was thus formed through the interaction of functional groups between the two. A new substance, amino acid-based NADESs, was synthesized.

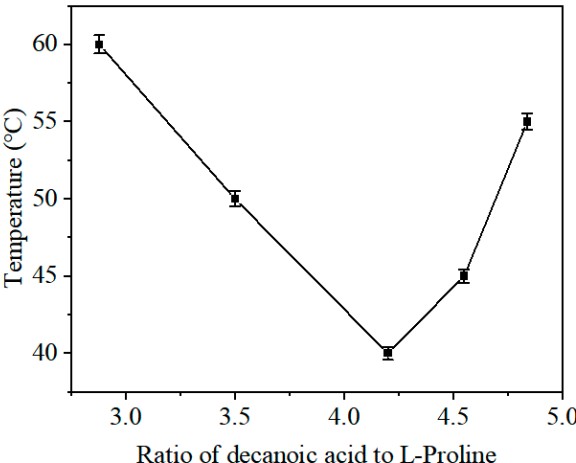

**Figure 1.** Synthesis temperature of amino acid-based NADESs at different molar ratios.

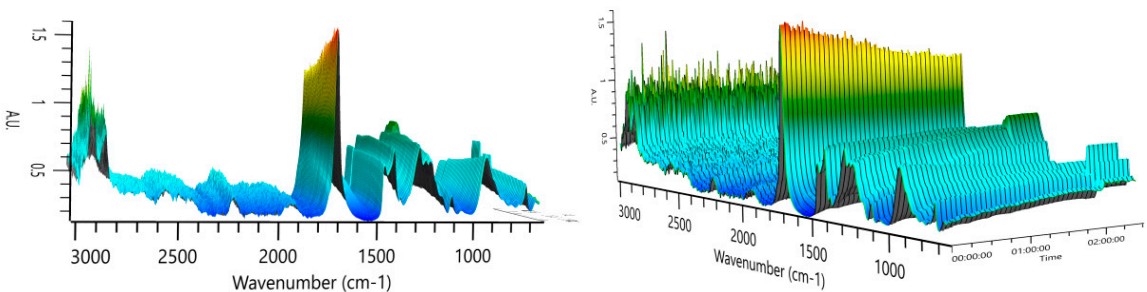

**Figure 2.** In situ IR spectra of preparation process of amino acid-based NADESs.

### 3.2. Properties of Natural Deep Eutectic Solvents

3.2.1. Thermostability

Thermal stability is a significant parameter related to the extractant with NADESs. The thermodynamic properties of synthesized NADESs were examined by TG-DTG analysis (Figure 3). Figure 3 shows that NADES had two absorption peaks at 28 and 200 °C, respectively. Sample mass started to decrease at 150 °C and reached its peak value at 200 °C according to the DTG curve. The NADES had a melting point of 28 °C and began to decompose at 150 until 255 °C, when it was fully decomposed. So, the NADES was stable at room temperature from 25 to 150 °C. This indicates that the prepared NADES had good thermal stability to meet phenol extraction.

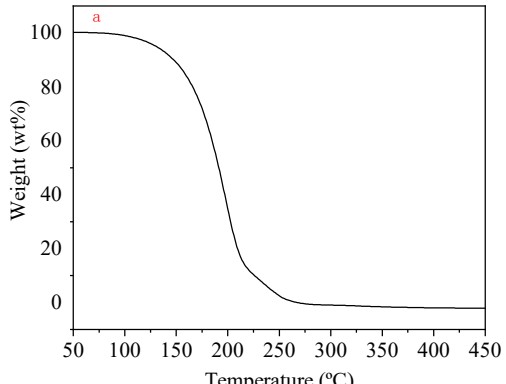
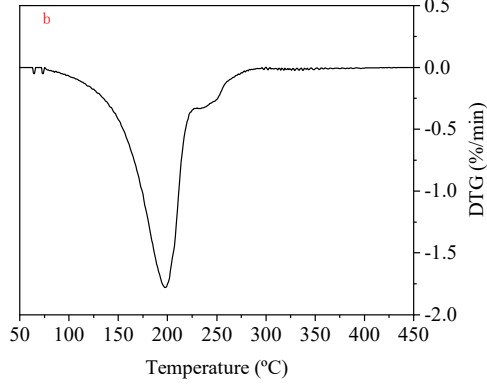

**Figure 3.** (**a**) TG (in nitrogen) curves of NADESs; (**b**) DTG (in nitrogen) curves of NADES (decanoic acid: L-Pro:4.2:1).

### 3.2.2. FT-IR and [1]H NMR Analysis

The surface chemical composition of the NADES was manifested with FT-IR and [1]H NMR, respectively (Figure 4a,b). In Figure 4a, the peaks at wavelengths of 2850.7, 2916.3, and 2951.0 cm$^{-1}$ denote the C−H stretching vibration for −CH$_2$− and −CH− of L-proline. The characteristic peaks 3136.2 and 1703.1 cm$^{-1}$ are attributed to O–H and C=O stretching vibration for the −COOH group in L-proline and decanoic acid, respectively. Particularly, the characteristic peak 927.7 cm$^{-1}$ is attributed to O−H bending vibration for the −COOH group. The characteristic N−H stretching peak appeared at 1610.5 and 1633.6 cm$^{-1}$ for C−NH. Combined with the structural formula of the precursor (Figure 4b, insets), 721.4 cm$^{-1}$ was the absorption peak of C−H bending vibration for −CH$_2$− in decanoic acid, and 1193.9 and 1220.9 cm$^{-1}$ were the absorption peaks of C−N stretching vibration in L-proline.

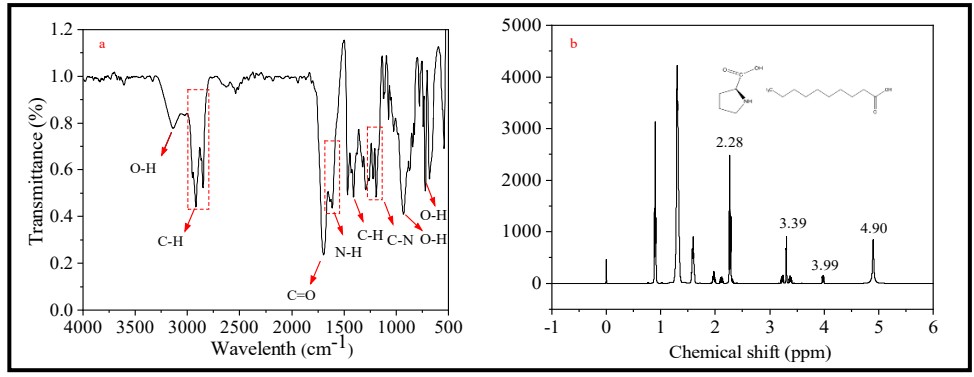

**Figure 4.** (**a**) FT-IR of NADES (decanoic acid: L-Pro:4.2:1); (**b**) 1 H NMR spectra of NADES (decanoic acid:L-proline 4.2:1). Insets: structural formula of L-Pro and decanoic acid.

Moreover, to verify the reaction between decanoic acid and L-proline, [1]H NMR results are shown in Figure 4b. The characteristic peak of −CH connected to −COOH in L-proline structure was at 3.99 ppm. Peaks at 2.28 and 3.39 ppm were assigned to −CH$_2$C=O on decanoic acid and −CH$_2$N− on L-proline, respectively. H on −NH− in L-proline is shown in 3.58 ppm. However, new peaks appeared at 4.90 ppm, indicating that a chemical reaction took place between the two raw materials, and the functional group changed. A new hydrogen structure was formed, which further proved the success of the synthesis of the amino acid-based NADESs.

### 3.2.3. pH

pH reflects the acidity of a solution, and it directly affects the extraction of phenolic compound. As shown in Table 1, the pH of NADESs is about 5–6 in aqueous environments and shows weak acidity. This provides a basis for the treatment of phenol wastewater.

**Table 1.** pH of NADESs with different molar ratios.

| Sample Number | Ratio of Decanoic Acid and L-Proline | pH | Acid/Base Properties |
|:---:|:---:|:---:|:---:|
| 1 | 4.2:1 | 5.67 | acidity |
| 2 | 4.55:1 | 5.42 | acidity |
| 3 | 3.5:1 | 5.58 | acidity |
| 4 | 4.84:1 | 5.24 | acidity |
| 5 | 2.88:1 | 5.86 | acidity |

### 3.2.4. Stability in the Presence of Water

Water content greatly influences the stability of NADES, and it can weaken the hydrogen-bond interaction between HBA and HBD, thus resulting in the dissociation

of NADES. It is important to moderate the water content of NADESs. The thermodynamic stability of NADESs with water content from 0 to 55% was determined by TG (Figure 5). The TG curves of decanoic acid/L-proline NADES maintained good stability when the content of water was not much more than 35% (V/V). This might have been due to the formation of a NADES hydrate, which was discussed in previous studies [41,42]. However, the hydrogen bond was gradually weakened with water added up to 45% (V/V). The smooth curve became steeper, and platforms even appeared. The NADES absolutely dissociated till water content was 55% (V/V), and transited from a united to a mixed aqueous solution. The result was consistent with that of Dai, who reported that 50% (V/V) was the critical value of water in NADESs [43].

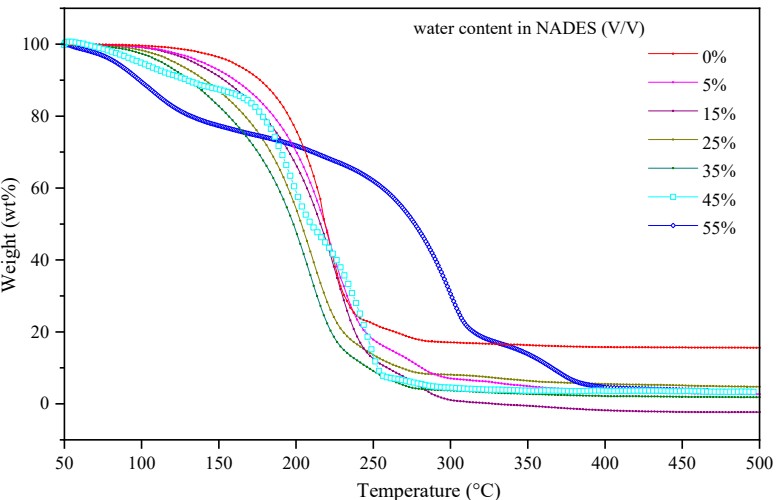

**Figure 5.** TG (in nitrogen) curves of NADESs with different water content levels.

### 3.3. Phenol Extraction with Natural Deep Eutectic Solvents

3.3.1. Standard Curve of Absorbance–Concentration

In this work, the maximal absorption wavelength of phenol by UV–vis absorption spectra was about 267 nm (Figure 6a,b). Simulated phenolic wastewater with concentrations of 16, 20, 40, 60, and 80 mg $L^{-1}$ were taken to measure UV–vis absorption spectra, respectively. The fitting-degree curve was drawn according to the five measured scatter points, and the fitting degree of the curve was calculated by computer as $R^2 = 0.9992347$. Its function expression $Y = 64.331X - 1.1872$ was also calculated. Maximal absorbance at the maximal absorption wavelength was 4.419. Substituted it into the function, the maximal phenol concentration measured by UV–vis absorption spectra was 283.09 mg $L^{-1}$ under the condition of no dilution.

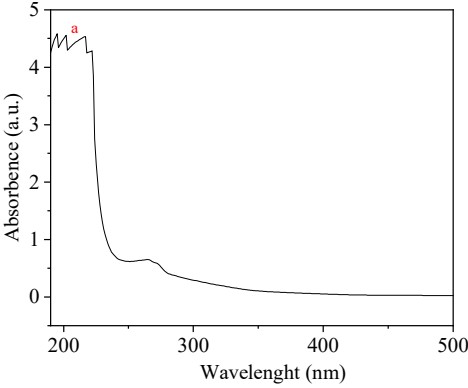 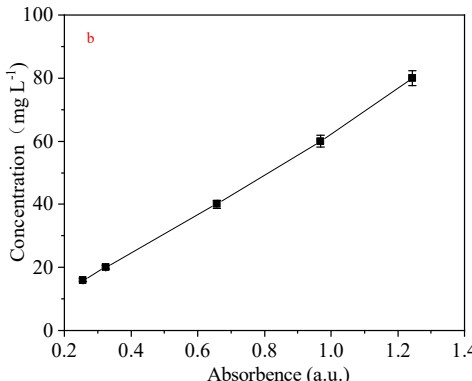

**Figure 6.** (**a**) UV–vis absorption spectra of 30 mg $L^{-1}$ phenol wastewater; (**b**) standard curve of absorbance–phenol concentration.

### 3.3.2. Effect of Precursor Molar Ratio on Phenol Extraction

After the reaction of NADESs and simulated phenol-containing wastewater, the two were quickly stratified. After standing for 10 min, the NADES layer was completely separated from the water layer, and residual phenol concentration was detected in the aquifer.

Figure 7 shows that phenol extraction rate varied with precursor molar ratio. The extraction rate was obviously related to precursor molar ratio. At 40 °C, reaction time was 30 min, and the concentration of phenol-containing wastewater was 100 mg L$^{-1}$. The maximal rate was 38% with a molar ratio of 3.5:1. In the range of 2.88–4.84 for the molar ratio of decanoic acid to L-proline, the extraction rate of phenol first increased and then decreased.

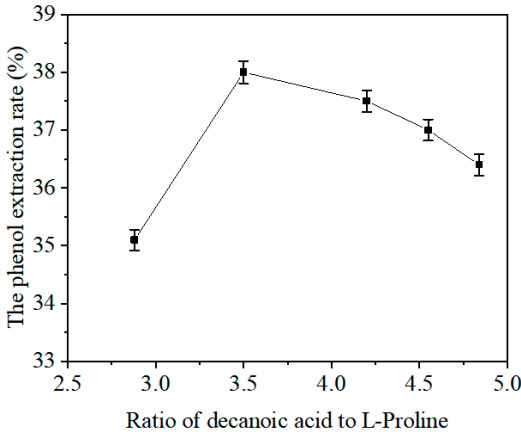

**Figure 7.** Effect of precursor molar ratio on phenol extraction rate.

To understand the extraction mechanism, in situ IR spectra were applied to monitor the extraction of phenol by NADES (Figure 8). In the formation of NADES between L-proline and decanoic acid with a molar ratio of 3.5:1, relative stable hydrogen bond H−O···H−N was observed, and the hydrogen bond was not so stable as that of 4.2:1, nor so loose as that of 2.88:1 or 4.84:1, which would afford potential electrons to react with electrophilic group on phenol. In the extraction of phenol (Figure 8a,b), the intensity of the hydrogen bond in NADES at 1700 cm$^{-1}$ greatly changed, as it decreased sharply. At the same time, the peak intensity at 1250 cm$^{-1}$ slightly increased with the addition of NADES to phenolic wastewater. This suggests the interaction of NADES and phenolic hydroxyl group by hydrogen bond. Moreover, the increase in N−H peak intensity at 1658 cm$^{-1}$ indicated that O−H in the phenolic compound combined with N−H in the NADES. Thus, phenolic compounds were extracted by NADES to achieve the treatment of phenolic wastewater (Figure 8c–e).

In subsequent experiments, a NADES with a molar ratio of 3.5:1 for decanoic acid and L-proline was selected.

Trioctylamine, kerosene, and tributyl phosphate are conventional organic solvents, and they have good phenol extraction efficiency. In order to evaluate the potential application of NADES, parallel experiments with conventional organic solvents were also carried out under the same extraction conditions. The phenol extraction rates of trioctylamine, kerosene, and tributyl phosphate were 45%, 34%, and 39%, respectively. Though the extraction efficiency of NADES (38%) was lower than that of trioctylamine and tributyl phosphate, the difference was small. As a new solvent, results demonstrated that NADES is a promising green extraction technology in the field of separation science, especially phenolic compounds in large-scale applications.

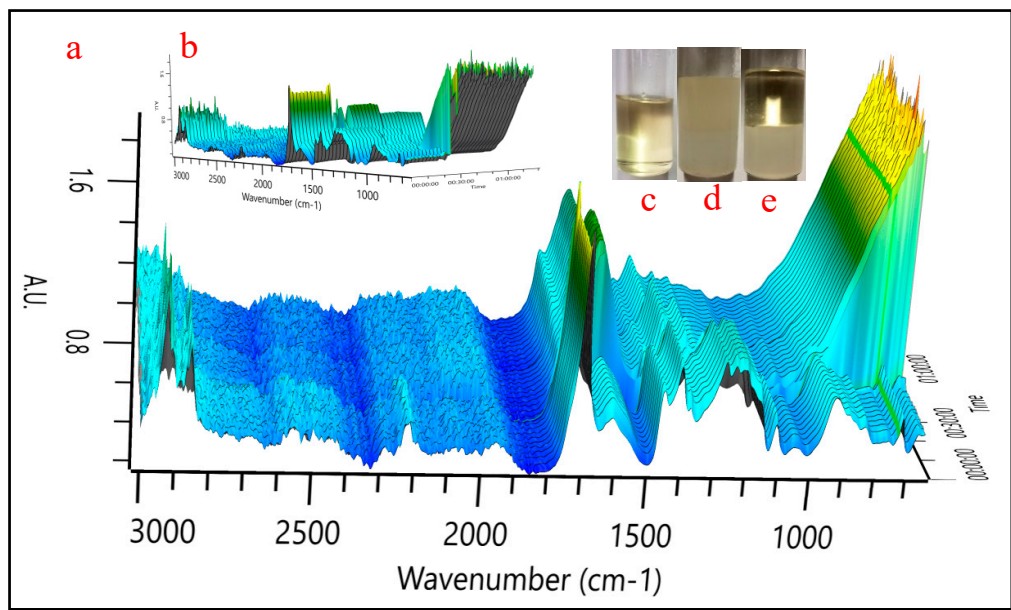

**Figure 8.** In situ IR spectra of NADESs and simulated phenol wastewater (0.5 and 100 mg L$^{-1}$). (**a**,**b**) Front and side views of in situ IR spectra; (**c**–**e**) diagram of phenol extraction process.

### 3.3.3. Effect of Phenol Concentration on Phenol Extraction

As shown in Figure 9, the effect of different phenol concentration on phenol extraction was investigated in a NADES system. Reaction conditions were the same as above. The extraction rate increased with the increase in phenol concentration. When the concentration of phenolic wastewater was greater than 300 mg L$^{-1}$, the extraction rate remained at about 53%. The extraction rate of phenol from NADES was less than 24% in simulated phenolic wastewater with a concentration of 0–10 mg L$^{-1}$, and the removal rate of phenol from simulated phenolic wastewater with a concentration of 0.5 mg L$^{-1}$ was 20%. Therefore, the minimal detection limit of phenol from the prepared NADES was 0.5 mg L$^{-1}$, which has a certain practical value. With the increase in concentration of wastewater containing phenol, the extraction rate of phenol increased. When the concentration of phenol in simulated wastewater reached 200 mg L$^{-1}$, the extraction rate of phenol fluctuated at 1% with the increase in the concentration of phenol. The maximal removal rate of phenol in NADES was 53% when the concentration of phenol in simulated wastewater was greater than 200 mg L$^{-1}$.

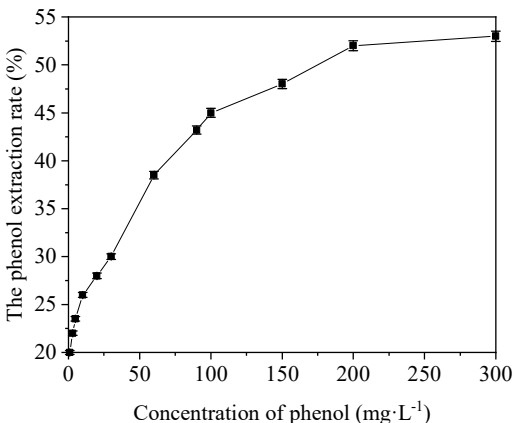

**Figure 9.** Effect of phenol concentration on phenol extraction rate.

### 3.3.4. Effect of Reaction Time on Phenol Extraction

From the start of the reaction to the maximal extraction rate, reaction time is essential for the practical application of NADESs. Generally, a short extraction time is beneficial to enhance processing capacity. In this study, NADES with a ratio of decanoic acid to L-proline of 3.5:1 was used to continuously stir and heat the simulated phenol-containing wastewater of 100 mg $L^{-1}$. At a specific time, phenol concentration was determined. The experimental results are shown in Figure 10.

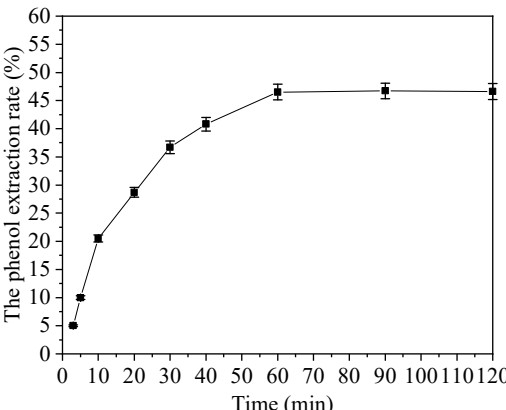

**Figure 10.** Effect of reaction time on phenol extraction rate.

When phenol concentration, reaction temperature, and precursor ratio were fixed, the extraction first increased and then tended to be stable with the extension of reaction time. After 15 min, the reaction was the most intense, and the intensity of the reaction decreased within 15–60 min. At 60 min, the reaction tended to be stable. At this time, the extraction rate of phenol was 46%, and fluctuation was no more than 1%. The extraction rate of phenol at 60 min was 8% higher than that at 30 min. Results showed that the reaction time greatly influenced the extraction rate of phenol. Therefore, the optimal reaction time is 60 min.

### 3.3.5. Effect of Temperature on Phenol Extraction

The effect of temperature on phenol extraction was investigated and is shown in Figure 11. The extraction rate of phenol were 53.0%, 62.0%, 58.2%, and 51.4% at the reaction temperature of 45, 50, 55, and 60 °C, respectively. There was a peak of 62.0% with reaction temperature of 50 °C, which then decreased slowly. The extraction rate of 40 °C was 17.9%, which was less than all the above. Reaction temperature greatly influenced the extraction rate of phenol, and the optimal reaction temperature was 50 °C.

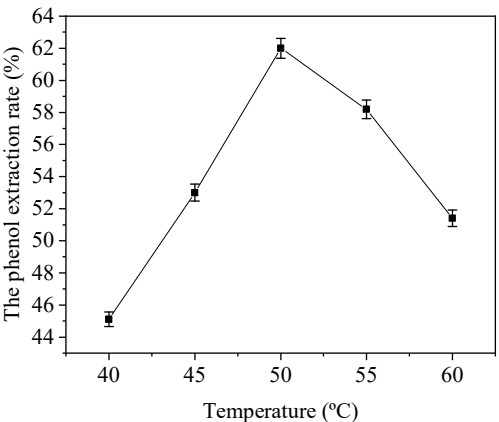

**Figure 11.** Effect of reaction temperature on phenol extraction rate.

### 3.3.6. Study on Recyclability of Natural Deep Eutectic Solvents

From an economic perspective, excellent extractant renewability plays a crucial role in practical applications, which suggests that the cost of wastewater treatment can be significantly reduced. In our study, chloroform was chosen to extract phenol in NADESs because phenol dissolves well in chloroform. The optimal NADES was mixed with 100 mg L$^{-1}$ phenolic water. After the NADES was completely stratified with phenolic water, the water phase was taken, and absorbance was tested. The phenol was removed by chloroform, and the 100 mg L$^{-1}$ simulated water was mixed with the NADES after dephenolic water.

Figure 12 shows that the removal rate of phenol changed little with the increase in cycle number. At the sixth extraction, the removal rate of phenol in NADES was still above 57%, and calculability was relatively high. For this, we refer to the available literature data. In research by Youan Ji et al. [44], biological reagents based on Brønsted acid–Lewis base interaction were used to separate phenol from the environment, Lastly, the extraction rate of phenol by 1,3-dimethylxanthine (DMX) was 66.6%. BRs were regenerated with an antiextraction method for four cycles without significant changes in separation efficiency and their properties. In addition, Yatong Zhang et al. [45] used biodiesel to extract phenol, and results showed that phenol extraction efficiency reached 98%. However, the regeneration ability of the extractant was not further investigated. Therefore, NADES as a green solvent has certain advantages in extracting phenol.

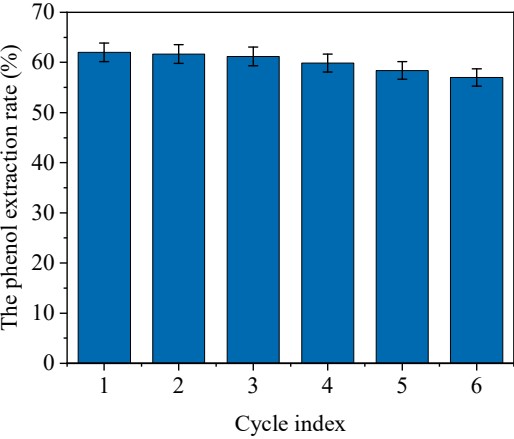

**Figure 12.** Effect of cycle number on phenol extraction rate.

### 4. Conclusions

In this work, NADESs with L-proline and decanoic acid at different molar ratios were synthesized and applied in the extraction of phenolic compounds from wastewater. Results showed that the green amino acid-based NADES had excellent physical and chemical properties, and most importantly, does not pollute the environment. In the separation application, the influencing factors for the extraction of phenol from wastewater were systematically studied. It could achieve high efficiency up to 62% by decanoic acid/L-proline NADES with a precursor molar ratio of 3.5 to 1 under a reaction time of 60 min and temperature of 50 °C. The NADES could be cycled for six times without a significant decrease in extraction capacity. The application of green NADESs in the treatment of hazardous substance combines cleaner production and sustainable development. Amino acid-based NADESs could be a substitute for the conventional organic solvent extraction of phenolic compounds from aqueous environments. They are also a promising green solvent for the field of separation science, which has great potential applications in industry.

**Author Contributions:** Conceptualization, Y.L. and F.H.; methodology, M.L.; software, M.L.; validation, M.L., H.R. and E.D.; formal analysis, Y.L.; investigation, F.H.; resources, E.D.; data curation, M.L.; writing—original draft preparation, M.L.; writing—review and editing, H.R.; visualization,

M.L.; supervision, H.R.; project administration, E.D.; funding acquisition, H.R. and E.D. All authors have read and agreed to the published version of the manuscript.

**Funding:** This research was funded by the National Science Foundation of China, Key R&D Program Projects of Hebei Province, Hebei Provincial Department of Science and Technology and Natural Science Foundation of Hebei Province, grant numbers 52004080, 20373703D, 19273711D (19943816G), and E2019208234.

**Institutional Review Board Statement:** Not applicable.

**Informed Consent Statement:** Not applicable.

**Data Availability Statement:** All data are available and can be shared upon request.

**Acknowledgments:** This work was supported by Innovation Funding Projects of Hebei University of Science and Technology (XJCXZZSS202107).

**Conflicts of Interest:** The authors declare no conflict of interest.

## Abbreviations

NADESs, natural deep eutectic solvents; FT-IR, Fourier transform infrared spectrometer; $^1$H NMR, hydrogen nuclear magnetic resonance spectrometer; TG-DTA, differential-thermo gravimetric analyzer; ILs, ionic liquids; HBDs, hydrogen-bond donors; HBAs, hydrogen-bond acceptors; RSM, response surface methodology; TrpBF$_4$/U, tryptophan fluoborate/urea; TBP, tributylphosphane; DEC, diethyl carbonate; UV–vis, ultraviolet–visible spectroscopy; L-Pro, L-proline.

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
