# Peer review of "Amino Acid-Based Natural Deep Eutectic Solvents for Extraction of Phenolic Compounds from Aqueous Environments"

_processes, doi:10.3390/pr9101716_

Round 1

Reviewer 1 Report

Some contents to be explained more clearly in the paper:

- Why were these specific DES components chosen for phenol extraction?

- In my opinion, the use of the term DES synthesis is exaggerated. Deep eutectic solvents (DESs) are a class of mixtures not new compound.

- The title of 2.3.1. “Synthesis of simulated wastewater” is s incorrect. No synthesis process is described in this subchapter. Analogously 3.1. Synthesis of natural deep eutectic solvents

- What happens to DES when the temperature exceeds 150C and 200C? What chemical changes occur?

- What kind of new structure was formed when decanoic acid and L-Proline were mixed? It is not clear.

- It would be clearer if the TGA results were presented together rather than separated. I suggest superimposing the TGA result for the sample "without water" on the DES results with water.

- I am concerned about the purity of reagents used to obtain DES. A purity of 98% could mean that the reagents contain more than 1% water.  Analysis of water content was not done. The compounds were used as received. In this system tests for samples containing 0.1% water are unreliable. In my opinion, the authors should use higher purity compounds or purify the reactants to obtain DES.

Reviewer 2 Report

The authors present an interesting work. The aim of the reported research is to extract phenols from wastewater using a natural deep eutectic solvent. In some places, the paper is not very clearly written, and the English tend to be poor. The manuscript requires a major revision before its acceptance for publication.

  1. On page 3, lines 99-104: What is HBD in the NADES preparation? It stated that L-proline and decanoic acid both are HBA.
  2. Section 2.2: It is not a synthesis of DES. Here, DES is the mixture of two precursors , so it should be the preparation of DES not synthesis. And what was the % level of the purity of the preparation DES?
  3. Section 3.1: What does it mean by temperature required for the synthesis of NADES? Is this temperature required to melt both compounds? or the melting temperature of prepared NADES? and how did the authors measured temperature?
  4. Correction in Figure 1 and Figure 7 (x-axis): Ration of decanoic acid to L-proline, It is not Ration, is is 'Ratio'.
  5. Did the authors measure the molar ratio of DES after its recycling? If yes, what is the molar ratio of HBA to HBD?
  6. How did the authors confirm the complete removal of phenol from NADES after the addition of chloroform? There may be a tough competition between chloroform and NADES for phenol. And did the authors measure the solubility of chloroform in NADES?
  7. I would suggest authors compare the present study results with available literature data with different solvents and efficacy of phenol removal percentage.

Round 2

Reviewer 1 Report

Accept in present form

Reviewer 2 Report

The authors responded well to all my concerns, hence I recommend this article for publication.